# Pulsed Radiofrequency Ablation for Orchialgia—A Literature Review

**DOI:** 10.3390/diagnostics12122965

**Published:** 2022-11-27

**Authors:** Meshari A. Alzahrani, Omar Safar, Muath Almurayyi, Abdulaziz Alahmadi, Abdulrahman M. Alahmadi, Muhannad Aljohani, Abdalah E. Almhmd, Khaled Nasser Almujel, Bader Alyousef, Hussam Bashraheel, Feras Badriq, Abdulaziz Almujaydil

**Affiliations:** 1Department of Urology, College of Medicine, Majmaah University, Al-Majmaah 11952, Saudi Arabia; 2Urology Department, Armed Forces Hospital Southern Region, Khamis Mushayt 62461, Saudi Arabia; 3Urology Department, King Khaled University Medical City, Abha 62529, Saudi Arabia; 4College of Medicine, Taibah University, Madinah 42353, Saudi Arabia; 5College of Medicine, Majmaah University, Al-Majmaah 11952, Saudi Arabia; 6College of Medicine, The Royal College of Surgeons in Ireland, D02 YN77 Dublin, Ireland; 7Department of Urology, King Fahad General Hospital, Jeddah 23325, Saudi Arabia; 8Division of Urology, Department of Surgery, East Jeddah General Hospital, Jeddah 22253, Saudi Arabia; 9Department of Urology, Ministry of Health, Riyadh 12613, Saudi Arabia

**Keywords:** chronic orchialgia, testicular pain, scrotal pain, pulsed radiofrequency

## Abstract

Pulsed radiofrequency, short bursts of radiofrequency energy, has been used by pain practitioners as a non- or minimally neurodestructive technique, an alternative to radiofrequency heat lesions. The clinical advantages and mechanisms of this treatment remain unclear. To review the current clinical implication of the pulsed radiofrequency technique for male patients with chronic scrotal pain. We systematically searched the English literature available at the EMBASE, MEDLINE/PubMed, Google Scholar, and Cochrane Library from inception to 22 November 2022. Only reports on a pulsed radiofrequency application on male patients with chronic scrotal pain were included. The final analysis yielded six reports on the clinical use of pulsed radiofrequency applications in male patients with chronic scrotal pain: six full publications, three case reports, one case series, one prospective uncontrolled pilot study, and one prospective randomized, controlled clinical trial. The accumulation of these data shows that using pulsed radiofrequency generates an increasing interest in pain physicians, radiologists, and urologists for managing chronic scrotal pain. No side effects related to the pulsed radiofrequency technique were reported to date. Further research on the clinical and biological effects is justified. Large sample sizes and randomized clinical trials are warranted.

## 1. Introduction

Worldwide, chronic pain is one of the major public health problems and is one of the leading causes of years lived with disability (YLDs) [1]. 

Chronic scrotal pain (CSP) “Chronic orchialgia” is defined as intermittent or constant, unilateral, or bilateral pain, localized to the scrotal structures, three months or longer in duration that significantly interferes with the daily activities of the patient and prompts him to seek medical attention [2].

CSP is frustrating for both the physician and the patient. Managing patients with chronic pain is difficult, frustrating, and time-consuming for the urologist and the patient. It is the cause of about 2.5% to 5% of all urology consultations and currently affects about 100,000 men in the United States each year [3].

Chronic orchialgia may include infection, tumor, inguinal hernia, hydrocele, spermatocele, varicocele, referred pain, trauma, and previous operations (e.g., herniorrhaphy, vasectomy, or other scrotal procedures). Nearly 25% of patients with chronic orchialgia have no apparent cause for the pain [4].

This condition has been referred to by many names, including chronic orchialgia, testicular pain syndrome, testalgia, chronic scrotal content pain, post-vasectomy orchialgia, post-vasectomy pain syndrome (PVPS), congestive epididymitis, and chronic testicular pain [5]. 

Chronic orchialgia is used frequently to describe what is better described as chronic scrotal content pain (CSCP), as the pain may involve the testicle only and/or the epididymis, para-testicular structures, and the spermatic cord [5]. For simple modification in this article, we will use the term chronic scrotal pain (CSP).

Multiple algorithms for diagnosing and treating CSP have been proposed, but none have been validated [5]. It is a complex condition to manage, and varied practices exist. Managing men with CSP typically involves a stepwise approach, starting with the least invasive options and moving to more invasive therapies. Possible three ways contributing to the etiology of pain in chronic orchialgia include [3]: The ilioinguinal nerve and the genital branch of the genitofemoral nerve innervate the testis’s major somatic sensory innervation. Genital pain, on the other hand, can be felt in any organ or tissue that shares the same L1–L2 or S2–S4 neural pathways as the scrotum or testicles. Orchialgia can be caused by any inflammatory, traumatizing, or infectious stimulus to the scrotal nerves. Testicular pain could be caused by low back pain or radiculitis, which affects the nerve roots from T10 to L1.Ureteral stones can produce testicular pain because the testis and the upper ureter have sensory fibers that use spinal cord segments T11 and T12.Dysfunction or spasms of the pelvic floor muscles can be linked to CPPS, although they can also occur in some chronic orchialgia patients.

There are many different modalities for CSP management ranging from conservative management as the first-line treatment in most cases, including heat or ice compression, scrotal elevation, antibiotics, analgesics, local and regional nerve blocks, pelvic floor physiotherapy, biofeedback, acupuncture, and psychotherapy for at least three months. Some cases may indicate surgical intervention when failed conservative management occurs; if the spermatic cord block is only 50% effective in reducing orchialgia, surgical intervention is indicated, including varicoceles or epididymectomy or microsurgical denervation of the spermatic cord (MDSC). Pulsed radiofrequency (PRF) therapy to denervate the spermatic cord has been tried with encouraging early results, although its final efficacy and duration are still uncertain [3].

Based on the best practice report on chronic scrotal pain from Canadian Urological Association, PRF denervation remains (Level 4 evidence, Grade C recommendation) [2].

The biological effects of the radiofrequency (RF) field have been investigated [6,7,8]. These studies led to a search for new applications to apply RF without diffuse tissue damage [8], like PRF, a new method of applying RF without raising the temperature. In PRF, the output of the generator is cyclically interrupted. The initial parameters were two cycles of 20 ms each of the active cycle. Nowadays, new parameters are used considering new biological discoveries that are continuously published [9]. Although the mechanism of action has not been completely elucidated, laboratory reports suggest a genuine neurobiological phenomenon altering pain signaling, which some have described as neuromodulators [10]. According to animal studies, PRF, a different way to deliver RF energy, has distinct biological effects on cell shape, synaptic transmission, and pain signaling, and is both temperature-independent and minimally destructive [10]. This review aims to explore the literature that discusses the application of PRF to patients with CSP.

## 2. Materials and Methods

Search strategy and information source: We searched seven databases via EMBASE, MEDLINE/PubMed, Google Scholar, and Cochrane Library. The included databases were searched from inception until 22 November 2022, including only English literature using search terms: pulsed radiofrequency AND chronic scrotal pain AND chronic testicular pain AND orchialgia.

Study selection and data extraction: Conducted in a standardized excel sheet that included the IDs, authors, URL, DOI, journal and database name, and abstracts to facilitate the screening process and hasten our study efficiently. Two reviewers independently screened each title/abstract of retrieved records and full texts of retrieved studies for possible inclusion discrepancies, which were resolved by senior authors who were consulted whenever needed. 

We extracted the following data: the surname of the first author, year and country of publication, study design, details about intervention (treatment protocol and device used), number of participants, baseline characteristics of participants, pain level pre- and post-PRF treatment, follow-up period and outcomes about efficacy and safety.

Eligibility criteria: The study inclusion and exclusion criteria for reviewed articles are summarized in (Table 1). We included primary studies with participants suffering from CSP. If that condition was defined of both origins, neuropathic and non-neuropathic, such as post-surgical pain or low back pain, we excluded such a condition. 

We included studies where PRF treatment was directed to the dorsal root ganglion (DRG) or peripheral nerves. If the study only reported results about efficacy and safety was not reported, we still included such a study to achieve a comprehensive evidence synthesis regarding efficacy. Each study was evaluated using the Joanna Briggs Institute Quality Assessment Tool (https://jbi.global/critical-appraisal-tools, accessed on 15 October 2022). Two reviewers extracted data independently using a pre-designed table to summarize study results, which included author, year, study design, and outcome measure information.

Outcome measures: Primary outcomes were pain intensity and serious adverse events (SAEs). Secondary outcomes for efficacy were any other pain-related outcomes and safety data, including non-serious adverse events and other complications regarding intervention.

Synthesis of results: We conducted a narrative and tabular synthesis of the results. We also analyzed the conclusiveness of efficacy and safety of the treatment in the abstracts of the included studies. For this reason, due to the heterogeneity of included studies, it was not possible to conduct a systematic analysis, even though we had planned to do it in our study protocol.

## 3. Results

We screened manuscripts in the full text, including six studies in this literature review. The total number of participants in these manuscripts was 55 patients, one prospective randomized double-blinded controlled trial including 30 patients, one prospective uncontrolled pilot study including nine patients, one case series of five patients, and three different case reports of three patients. Eight patients have been excluded, four were lost to follow-up, and four did not match the inclusion criteria. In these studies, a total of 47 patients were subjected to RFA for the treatment of CSP. Detailed information about inclusion and exclusion criteria and baseline characteristics of included participants are listed in (Table 1). Studies and patient characteristics, age (mean, SD), pain duration, follow-up, and main outcome are listed in Table 2.

Only five patients had no treatment before RPF, 42 patients underwent conservative management, including antibiotics, analgesics, anti-inflammatory agents, or antidepressant drugs, and five patients had transcutaneous electrical nerve stimulation (TENS). Moreover, all the patients underwent diagnostic cord blockage before RPF. Forty-six patients had PRF at the spermatic cord in the setting of 2-Hz frequency, 20-ms pulses in one cycle, 120-s duration, and 42 °C temperature, except one patient who had a duration of RPF for 180 s. However, one patient received PRF at the dorsal root ganglia in the setting of 2 Hz, 8 ms pulse duration, max 45 V, and max 42 °C for 8 min (Table 3). The duration of the procedure was reported with a mean (average) of 5.1 min, the short duration was 2 min, and the longest was 8 min (Table 4). Pain scale assessment before and after pulsed radiofrequency is detailed in Table 5. Follow-up protocol, follow-up duration in months reported recurrence, and complications are in Table 6.

### 3.1. Clinical Reports on Pulsed Radiofrequency and CSP

Prospective studies: We found one prospective randomized, double-blind, controlled clinical trial including 60 patients with three months follow-ups. The percentage of patients that showed >50% reduction of their visual analog scale (VAS) pain score and the percentage of patients that did not require additional analgesic drugs were assessed. In 50% of patients that did not require analgesic drugs, the mean post-procedural VAS pain reduction was significant (*p* = 0.001). 

In one prospective uncontrolled pilot study, nine patients were evaluated; four had complete resolution of pain, while one had partial pain relief. Three patients experienced no change, and one reported his pain worsening. McGill Pain Questionnaire and VAS were used with a mean long-term follow-up of 9.6 months (range 3–14 months). Both studies showed that no side effects were reported during the treatment and follow-up periods. 

Case Series and Reports: In one case series including five patients, the median VASs before intervention was 9, and the mean VAS, 3 weeks after the procedure was 1 showing a significant reduction of VAS scores (*p* < 0.05). The mean follow-up period was 20 (±) 2.5 weeks. None of the patients needed any analgesics after the procedure or during the follow-up. However, Scrotal Doppler ultrasonography was performed to evaluate the volume changes of the testis in the 12th week, and no complications were reported, and no recurrence was noted.

Two patients reported an immediate decrease in pain of >70% and 90%, respectively, and one patient showed a complete resolution of pain after a 7-month follow-up period. In addition, three case reports were found, including five patients. Two patients were excluded because they did not meet our inclusion criteria, i.e., female. Moreover, all these reports used a visual analog scale (VAS) pain score.

### 3.2. Diagnostic Nerve Block 

It is important to determine whether the pain is referred to or originated from the testes to treat chronic orchialgia. For differential diagnosis, diagnostic cord blockade must be performed. If the pain relief occurs several hours after the diagnostic cord blockade in a patient with chronic orchialgia, the pain originates from the testes [17,18,19,20].

Cohen and Foster studied all nerve blocks performed at the first visit according to standard practice and procedures [21,22]. Furthermore, before proceeding with PRF, all patients reported a recurrence of their pain to baseline at the next visit. All patients applied diagnostic nerve block with a resolution of the pain several hours after the blockade (Table 3).

### 3.3. Assessment Tool 

All reviewed studies used the visual analog scale (VAS), a 10-point (0 = no pain to 10 = worst pain) [23] to evaluate pain to provide overall intensity scores. One study used the perceived global effect (GPE) along with VAS. The GPE is a scale that asks the patient to rate, on a numerical scale, how much their condition has improved or deteriorated since some predefined time point [24]. The GPE was assessed by a 7-point Likert-like verbal rating scale: extremely dissatisfied = 1, dissatisfied = 2, somewhat dissatisfied = 3, undecided = 4, somewhat satisfied = 5, satisfied = 6, and extremely satisfied = 7.

One study used the short-form McGill Pain Questionnaire (SF-MPQ) [25], which includes the Present Pain Intensity (PPI) index of the standard MPQ and a VAS. The assessment tools per study and the outcome of PRF treatment are summarized in Table 5.

### 3.4. Proposed Mechanism of Action of PRF

Sluijter used this machine in early 1996 to conduct preliminary clinical trials and wrote the first report on the clinical effects of PRF on the dorsal root ganglia in 1998 [26].

PRF uses radiofrequency current in short (20 ms), high-voltage bursts; the “silent” phase (480 ms) of PRF allows time for heat elimination, generally keeping the target tissue below 42 °C within the range requisite for tissue destruction, as demonstrated by Cosman and Cosman [27]. It does not cause thermal lesions; data from an in vitro study showed the possibility of tissue destruction in PRF when using PRF electrodes at 60 °C or higher [27]. 

PRF ablation allows heat dissipation and is usually applied up to 40–42 °C by applying “pulses” of radiofrequency energy with intervening pauses [28,29]. Possible mechanisms of action include structural cellular damage, neuronal activation, and alterations in gene expression [30]. 

PRF has successfully treated painful metastatic tumors involving the brachial plexus and spinal facet pain, trigeminal neuralgia, chronic shoulder pain, cervical radicular pain [28], and refractory pudendal neuralgia [15,31].

## 4. Discussion

The first series of applied PRF has been published on the subject involving four patients with sciatica or post-thoracotomy pain, which purportedly works through the induction of an electromagnetic field [32]. It is crucial to determine whether the pain is referred to or originated from the testes to treat chronic orchialgia. For differential diagnosis, diagnostic cord blockade must be performed. If the pain relief occurs several hours after the diagnostic cord blockade in a patient with chronic orchialgia, the pain originates from the testes [17,18,19,20]. We included in this study all the patients who underwent diagnostic cord blockade and presented with a resolution of the pain several hours after the blockade.

The total number of participants was 47 patients. Age (mean, SD), pain duration, and follow-up were also summarized in (Table 2). One study showed no statistically significant between demographic data or patient characteristics among RFP vs Sham group, as reported by Hetta et al. [16].

Treatment options for chronic orchialgia include nonsurgical (e.g., NSAID, alfa-adrenergic antagonists, tricyclic antidepressants, transcutaneous electrical nerve stimulation (TENS), and pulsed radiofrequency) and surgical (e.g., pelvic plexus blockade under trans-rectal ultrasound (TRUS) guidance, laparoscopic denervation of the spermatic cord, microsurgical denervation of the spermatic cord, microsurgical testicular denervation, epididymectomy, and orchiectomy) procedures [13,33,34]. Microsurgical denervation of the spermatic cord for the treatment of chronic orchialgia is one of the minimally invasive approaches [20,35] with a reported success rate between 70% complete remission and 20% partial relief from pain [20]. However, it may cause possible distortion of anatomy and requires hospital admission. 

The perspective on treating chronic orchialgia clearly shows that the main advantage of PRF from surgical denervation procedures is that this is a minimally invasive technique with no adverse effects. PRF does not damage the nerve. The absence of neuroma formation may explain its increased success rate [13]. Cohen and Foster performed PRF procedures on three different peripheral nerves—ilioinguinal, iliohypogastric, and genital branches of genitofemoral—and showed excellent pain relief at six months. However, using PRF on nerve roots requires fluoroscopic imaging and precise needle positioning and is usually preceded by diagnostic nerve root blocks [11]. In this situation, the main disadvantage of the use of PRF is it being operated dependently. 

The procedure duration mainly ranged from 2–3 min without any medication or analgesic needed post the procedure. Moreover, in Misra et al.’s paper PRF was used (model RFG-3C, Radionics) with the following settings: 2-Hz frequency, 20-ms pulse, and 120-s cycles for a total of 8 min; the output voltage was adjusted for the temperature not to exceed 42 °C [12]. In contrast, Terkawi et al. mentioned that PRF of 42 °C was applied for 120 s (NeuroTherm NT 1100, Wilmington, MA, USA) was used [15]. 

Interestingly, all the patients reported an excellent outcome with the variation of the analgesia type and doses, technique, and procedure duration of RPF considering patients’ demographic factors, i.e., BMI and subjective pain before and after the procedure. However, conducting randomized placebo-controlled trials in interventional pain management has methodological and ethical limitations [10]. Therefore, a placebo effect cannot be ruled out, and more extended follow-up data are needed to confirm long-term efficacy [12]. Chronic scrotal pain management is crucial for both patient and physician, as there is no established effective treatment regimen nor a recognized and accepted standard protocol for evaluation.

A review of prospective longitudinal studies concludes that PRF exhibits a high safety margin in various conditions, including discogenic pain, chronic inguinal herniorrhaphy pain, and chronic testicular pain. They demonstrate the use of PRF in various indications and, in some instances, showed positive results where conventional treatments, on average, failed. However, these studies are subjected to a great degree of publication bias. Its efficacy, however, needs to be verified against identical control subjects. There also exist unanswered questions regarding the effective “PRF dose” based on voltage settings and duration of PRF treatment which require further clinical studies to substantiate [36]. 

Mean VASs before and three weeks after the procedure were 9 and 1, respectively. The differences between pre–PRF procedure and post–PRF procedure VAS scores were statistically significant (*p* < 0.05) [13]. Hetta et al. reported that VAS pain scores significantly reduced the mean overall VAS pain score over time between the groups (*p* < 0.01). Moreover, there was a significant reduction in the mean VAS score (between groups) at all of the measured time points (2-, 4-, 6-, 8-, and 12 weeks post-procedure) (*p* = 0.001) in the PRF group [16].

This study is not without limitations. The number of patients in this review is insufficient to draw firm conclusions, but the results are encouraging. The majority of reported studies’ follow-up was relatively short, which may affect the review’s depth and rigor. 

The strength of this review is that it is considered to provide the first insight in the English literature on available information and outcomes regarding the effect of RFA on chronic orchialgia. 

## 5. Conclusions

Pulsed radiofrequency (PRF) is promising and encourages a minimally invasive treatment for chronic scrotal pain with a high safety and tolerability profile. However, more extensive studies are needed with extended follow-up before any evidence-based recommendations can be made.

## Figures and Tables

**Table 1 diagnostics-12-02965-t001:** Inclusion and exclusion criteria of reviewed studies.

Inclusion Criteria
Male gender
Chronic orchialgia or chronic scrotal pain (CSP) that has been persistent for more than three months
The patient complained or was diagnosed with CSP
The patient undergoes pulsed radio-frequency therapy for orchialgia
Previous patient with hydrocelectomy or varicocelectomy or radical orchiectomy or vasectomy or vasectomy reversal surgery with CSP, (chronic post-surgical chronic scrotal pain), (post-surgical orchialgia), chronic post-groin surgery orchialgia (ilioinguinal and/or the genital branch of the genitofemoral nerves injury during surgical dissections)
Post-vasectomy pain
Pain post testicular torsion, pain with varicocele or hydrocele or testicular or epididymal or spermatocele without evidence of infectious causes.
Patients who showed more than a 50% reduction of their VAS pain score in response to diagnostic spermatic cord block Pain intensity ≥ 5 on the visual analog scale (VAS), pain that lasted for more than 3 months after groin surgeries, failed conservative treatment with nonsteroidal anti-inflammatory drugs (NSAIDs) and showed more than 50% reduction of their orchialgia on the VAS for at least 6 h following spermatic cord block with 6 mL of lidocaine 2%.
**Exclusion criteria**
Female gender
Post herniorrhaphy pain syndrome or inguinodynia, Chronic ilioinguinal neuralgia after inguinal herniorrhaphy, post-traumatic pain, post-abdominal surgery pain, post-abdominal radiation pain, post-cardiac catheterization (inguinal approach) pain, genitofemoral nerve (GFN) neuropathic pain after live laparoscopic donor nephrectomy.
Patient with the possibility of referred pain, psychiatric problems, malingering associated with chronic pelvic pain or ureteral stone or inguinal hernia or aortic or common iliac artery aneurysms, or lower back disorders.
Previous hernia, nerve entrapment, chronic inguinal neuropathic pain, chronic inguinal neuralgia, Ilioinguinal neuropathy post pelvic surgery
Patients with inflammatory causes of scrotal pain, e.g., groin infection, epididymitis, orchitis, and infected hydrocele.
Patients with coagulopathy, hypertension, ischemic heart disease
Patients allergic to local anesthetics like lidocaine and concomitant use of NSAIDs or monoamine oxidase inhibitors
The patient who lost follow-up from the study.

**Table 2 diagnostics-12-02965-t002:** Summary of study and patient characteristics who undergo pulsed radio-frequency therapy for orchialgia.

Author/Year	Country	Study Type/Methods	No. of Patients *	Age Range (Mean, SD)	Presentation and Laterality	Pain Duration (Months)	Follow Up	Outcome
Cohen and Foster (2003) [11]	USA	Case report	1	30	Aching and precipitated by activity. Location: Unilateral (right Side) Risk factor: post vasectomy.	60 months	6-months	Patient-reported >90% pain reduction. No complications were reported.
Misra (2009) [12]	UK	Prospective uncontrolled pilot study	9	Mean: 32–65 SD: 10	Aching, shooting gnawing Sickening Location: Unilateral: 6 Bilateral: 3	Median 36 months (range 6–120).	6 months (Range 3–14 months)	Four out of nine patients had complete resolution and one had partial pain relief. One patient reported worse pain Three patients didn’t experience any change. One patient lost follow-up. No complications were observed immediately or during the follow-up period
Basal (2012) [13]	Turkey	Case series	5	Mean: 29	All patients presented with chronic orchialgia Location: Unilateral: 4 (3 left sides, 1 right side) (Bilateral: 1)	7.8 months (5–12 months)	20 ± 2.5 weeks	None of the patients needed any analgesics after the procedure or during the follow-up period. No complications occurred after the procedure.
Hofmeester (2013) [14]	Netherlands	Case report	1	13	Pain arose in the right lower quadrant of his abdomen and radiated to the inside of his right upper leg and his right testicle. It was sharp, stinging. It was aggravated by walking, standing, and exercise, and only alleviated by bed rest. Location: Unilateral (Right)	2 years history of right-sided orchialgia	12 months	This led to an immediate and lasting decrease in pain of more than 70% as reported by the patient. No complications were reported.
Terkawi (2014) [15]	Saudi Arabia	Case report	1	17	Sudden attacks of severe scrotal pain on average once per month which had failed all conservative treatments The pain attacks were associated occasionally with syncopal attacks. Attacks were associated also with the frequency and dripping of urine. The patient could not maintain an erection because of the severe associated testicular pain. Trigger: cold weather Location: Bilateral	Orchialgia had started almost 6 years ago but had become more severe over the last 2 years.	7 months	After a 7-month follow-up period, the patient reported satisfactory analgesia with a VAS of 0/10 There were no complications, specifically no testicular atrophy or sexual problems, and the cremaster reflex was present bilaterally.
Hetta 2018 [16]	Egypt	Prospective randomized, double-blind, sham-controlled, clinical trial,	60, PRF group (n = 30), Sham group (n = 30).	PRF group: 35.8 ± 9.7 Sham group: 37.7 ± 10.4	pain intensity ≥ 5 on VAS	3 months post-groin surgery	3 months	>50% reduction of their VAS pain score was 80% (24/30) in the PRF group versus 23.33% (7/30) in the sham group percentage of patients that did not require analgesic drugs was 50% (15/30) in the PRF group versus 3.3% (1/30) in the sham group.

SD: standard deviation; PRF: pulsed radiofrequency; VAS: visual analogue scale. * Case selections based on inclusion and exclusion criteria.

**Table 3 diagnostics-12-02965-t003:** Parameters setting of pulsed radio-frequency therapy for orchialgia.

Author/Year	Interventions before PRF Treatment	Diagnostic Spermatic Cord Block Method before PRF	PRF Location	PRF Settings
Cohen and Foster (2003) [11]	Previous unsuccessful interventions included: (1) multiple medication trials (failed conservative treatment with antibiotics and nonsteroidal anti-inflammatory analgesics) (2) an ilioinguinal/iliohypogastric nerve block.	Ilioinguinal (T12-L1) nerve, iliohypogastric (T12-L1), and genital branch of the genitofemoral nerve (L1–L2) blocks were performed.	right-sided L1 dorsal root ganglion pulsed RF procedure was attempted (at the next visit) PRF procedures on three different peripherals. nerves—ilioinguinal, iliohypogastric, and genital branch of genitofemoral	2-Hz frequency, 20-ms pulses in a 1-s cycle, 120-s duration, and 42 °C temperature. Impedance ranged between 200 and 450 Ohms.
Misra (2009) [12]	Four out of nine patients had previous treatment as follows: (1) Transcutaneous nerve stimulation, ilioinguinal nerve block—no benefit (2) Ilioinguinal nerve block—worse (3) Obturator nerve block, Ilioinguinal nerve block, Injection of pubic tubercle—temporary relief (4) Ilioinguinal and genitofemoral nerve block—50% relief	applied diagnostic cord blockade to all of the patients, who were presented with a resolution of the pain several hours after the blockade	Spermatic cord	RF probe placed percutaneously into the SC was used to deliver four 120-s cycles of 20-millisecond pulses at 2 Hz.
Basal (2012) [13]	All patients had failed conservative treatment with antibiotic therapy, analgesics, anti-inflammatory agents, or antidepressant drugs.	blockade of the spermatic cord with 0.5% bupivacaine was applied 1 week before the procedure to those patients who did not respond to medical treatment	Spermatic cord	- 2 × 20 ms per second with a generator output (Cosman RFG-1A Lesion Generator, Gulhane Military Medical Academy, Ankara, Turkey) of 45 V for a duration of 3 min at 42 uC. The impedance range was 200–450 V. - outpatient procedure
Hofmeester (2013) [14]	Pharmacological conservative treatments including acetaminophen, NSAIDs, opiates, and gabapentin, combined with rest did not affect the pain. Transcutaneous electrical nerve stimulation (TENS) reduced the pain to some extent.	With fluoroscopy guidance, each of the dorsal root ganglia of thoracic 12, lumbar 1, and lumbar 2 (Th12, L1, and L2) on the right side was infiltrated with 1 mL of levobupivacaine 0.25%. This resulted in an excellent response lasting 6 h.	Dorsal root ganglia of thoracic 12, lumbar 1, and lumbar 2 (Th12, L1, and L2) on the right side.	(8 min, 2 Hz, 8 ms pulse duration, max 45 V, max 42 °C).
Terkawi (2014) [15]	Pharmacological conservative treatments (NSAID, Pregabalin, and Tramadol).	With the patient supine and using an aseptic technique, a high-frequency probe (3–12 MHz), (CX50 Philips, Bothell, WA, USA) was used to perform a selective block of the genital branch of the genitofemoral nerve. The area containing the spermatic cord was zoomed by the ultrasound for better visualization of the genital branch of the genitofemoral nerve, which was located lateral to the deferens duct. Local anesthetic injection was performed between the internal and the cremaster fasciae using a 22-gauge insulated needle (in-plane technique), using 20 mg of methylprednisolone and 5 mL of 2.5% bupivacaine bilaterally. The patient reported immediate pain relief that lasted for 6 weeks.	Genital branch of the genitofemoral nerve, which was located lateral to the deferens duct. Ultrasound-guided, pulsed radiofrequency ablation of the genital branch of the genitofemoral nerve.	The patient was counseled on the risks and benefits of thermal ablation using the same ultrasound-guided technique and opted for treatment. For the radiofrequency ablation, a 50 mm, 22 Gauge SMK needle and a 50 mm neuropile connected to the radiofrequency generator (NeuroTherm NT 1100, Wilmington, MA, USA) were used. After sensory and motor stimulation (the latter by watching for cremaster muscle contraction), a PRF of 42 °C was applied for 120 s.
Hetta 2018 [16]	Failed conservative treatment with nonsteroidal anti-inflammatory drugs (NSAIDs)	spermatic cord block with 6 mL of lidocaine 2%.	PRF was applied to the ilioinguinal nerve and the genital branch of the genitofemoral nerve	voltage output: 40–60 V; 2 Hz frequency; 20 ms pulses in a one-second cycle, 120-s duration per cycle, and 42 °C plateau temperature. PRF was applied for 4 cycles.

PRF; pulsed radiofrequency, RF; radiofrequency.

**Table 4 diagnostics-12-02965-t004:** Outcome parameters of pulsed radio-frequency therapy for orchialgia.

Author/Year	Need of Local Anesthetic Injection before PRF	Procedure Duration	Any Analgesic or Medication Needed Post PRF	Improvement (Perceptible Difference)	Resume Normal Activities	Main Outcome
Cohen and Foster 2003 [11]	Diagnostic nerve block	120-s (2 min)	NR	Complete resolution After a 6-month follow-up visit	NR	All 3 patients reported complete pain relief
Misra (2009) [12]	No local anesthesia	120 s/cycle for 4 cycles (8 min)	NR	within 2 weeks	They were all able to resume normal activities.	- Four patients described near complete pain relief. - One patient reported partial relief. - Three patients were unaffected. - One reported that he was significantly worse after the procedure.
Basal (2012) [13]	Diagnostic nerve block	3 min	Not needed	The mean follow-up period was 20 +− 2.5 weeks.	Just after the procedure	Effective and safe procedure for all patients
Hofmeester (2013) [14]	Diagnostic nerve block	8 min	NR	immediate decrease in pain of more than 70%	Immediate	- Pain relief - the patient was very pleased
Terkawi (2014) [15]	Diagnostic nerve block	120-s (2 min)	Not used	After 7m complete pain relief.	NR	Seven-month follow-up revealed complete Address for correspondence: satisfactory analgesia.
Hetta 2018 [16]	Spermatic cord block with 6 mL of lidocaine 2%.	120 s/cycle for 4 cycles (8 min)	50% of patients did not require analgesic drugs	The mean VAS pain score reduced from 5.97 to 2.07 by 2 weeks	NR	80% of patients showed >50% reduction in their VAS pain score

NR; not reported, VAS; visual analogue scale.

**Table 5 diagnostics-12-02965-t005:** Pain scale assessment before and after pulsed radio-frequency therapy for orchialgia.

Author/Year	Standardized Assessment Tools	Score before Pulsed RF	Score after PRF	*p*-Value
Cohen and Foster 2003 [11]	VAS	Median: 4.33 (4–5) Range: 4–5	Complete resolution	NR
Misra (2009) [12]	McGill Pain Questionnaire, And VAS	Median: 7.1 Range: 4.3–8.7	4.2 (0–8.6)	NR
Basal (2012) [13]	VAS	Median: 9	Mean: 1	≤0.05
Hofmeester (2013) [14]	VAS	Mean: 8.3	(Mean 1.6)	NR
Terkawi (2014) [15]	VAS	VAS: 10/10	VAS: 0/10	NR
Hetta 2018 [16]	VAS GPE	VAS: 5.90	2.40 (after 12 weeks)	0.001

PRF; pulsed radiofrequency, VAS; Visual Analogue Scale, GPE; Global Perceived Effect, NR; not reported.

**Table 6 diagnostics-12-02965-t006:** Follow-up protocol after pulsed radio-frequency therapy for orchialgia.

Author/Year	Lost Follow-Up Cases	Follow-Up Duration (Months)	Any Analgesic or Medication Needed during the Follow-Up Period	Disease Recurrence	Scrotal Doppler Ultrasonography	Complications
Cohen and Foster 2003 [11]	0	6 months	NR	No recurrence	Not done	No complications
Misra (2009) [12]	One patient	Mean 9.6 months. range of 3–14 months	NR	One patient reported some recurrence of pain on the same side a year after treatment, but the pain was much less intense.	Not done	NR
Basal (2012) [13]	0	Mean 20 ± 2.5 weeks. Shortest: 17 weeks Longest: 23 weeks	NR	No recurrence	At 12 weeks post PRF: No pathological changes No Changes in Testicular Volume at 3 months	NR
Hofmeester (2013) [14]	0	12 months	NR	No recurrence	Not done	No complications
Terkawi (2014) [15]	0	7 months	NR	No recurrence	Not done	No complications
Hetta 2018 [16]	35 patients done RFA, 2 protocol violations, 3 lost of follow-up. Remaining 30 patients	3 months	Duloxetine as the first line started at 30 mg and increased to 60 mg. Pregabalin as the second line started at 50 mg and increased according to efficacy. Tramadol as the third line started at 50 mg and changed according to efficacy. 15 patients required Analgesia	NR	Not done	NR

NR; not reported, VAS; Visual Analogue Scale, GPE; Global Perceived Effect.

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
