# Peer review of "Pulsed Radiofrequency Ablation for Orchialgia—A Literature Review"

_diagnostics, 2022, doi:10.3390/diagnostics12122965_

Round 1

Reviewer 1 Report

Elaboration will be interesting for clinicians as the problem of orchialgia is not well known in the same s treatment. It is described in well-English. Some improvements are suggested.

1.Line88-give more details

2. Please extend the study length till November

3. Provide information what are other treatments used for orchialgia

4. Sometimes patients have difficulty with the definition of where is the pain localised, or the pain place is dynamically changing then maybe the problem comes from pelvic floor muscle tension. It should be also mentioned and discussed differences in diagnosis and treatment (physiotherapy).

4. Line 309 please clearly describe the study limitations

Author Response

We want to thank the reviewers for their valuable comments; changes have been highlighted by a yellow mark in the main manuscript; here is the response as follows:

1.Line88-give more details

Response: detailed provided

2. Please extend the study length till November

Response: done

3. Provide information what are other treatments used for orchialgia

Response: we added a few details for orchialgia management

4. Sometimes patients have difficulty with the definition of where is the pain localised, or the pain place is dynamically changing then maybe the problem comes from pelvic floor muscle tension. It should be also mentioned and discussed differences in diagnosis and treatment (physiotherapy).

Response: we added a few details about the mechanism of orchialgia pain

  1. Line 309, please clearly describe the study limitations

Response: we added more information about the study limitations

Reviewer 2 Report

Chronic scrotal content pain (CSP) is a common, yet poorly understood condition that significantly impacts quality of life.

The manuscript is unique of its kind with reference to chronic scrotal pain (CSP), and what I understood was this manuscript data was generated based on published literature available in the public domain. In table 2 the author mentioned that In USA only 1 patient was available or was diagnosed. 

The evidence currently available remains rare and of low quality, making it difficult to strongly recommend individual treatment options. However, multimodal treatment modalities using physical therapy and psychotherapy may help patients and provide useful tools for coping with this condition. Pulsed Radiofrequency Ablation (PRF) technology represents a promising step toward treating complicated pain conditions. As the evidence in support of PRF accumulates, it is likely that its potential to be applied more broadly will also increase.

Author Response

We want to thank the reviewers for their valuable comments; changes have been highlighted by a yellow mark in the main manuscript; here is the response as follows:

Response:

We want to thank the reviewer for the feedback. CSP is a devastating disease affecting patient life quality and health.  Indeed, the role of RFA in CSP is promising and encouraging, we highlight some points to improve the manuscript entity in the yellow mark, and we add a few comments for further recommendations to improve RFA utility for the patient with CSP. We hope this review encourages urology and pain management researchers to do more clinical studies exploring the effect of RFA in managing chronic arthralgia. 
